# Mapping Kenyan Grassland Heights Across Large Spatial Scales with Combined Optical and Radar Satellite Imagery

**Olivia S.B. Spagnuolo [1],\*, Julie C. Jarvey [1], Michael J. Battaglia [2], Zachary M. Laubach [1,2], Mary Ellen Miller [2]** **, Kay E. Holekamp [1] and Laura L. Bourgeau-Chavez [2]**

[1]  Michigan State University, East Lansing, MI 48823, USA; jarveyju@msu.edu (J.C.J.); zachary.laubach@colorado.edu (Z.M.L.); holekamp@msu.edu (K.E.H.)
[2]  Michigan Tech Research Institute, Michigan Technological University, Ann Arbor, MI 48105, USA; mjbattag@mtu.edu (M.J.B.); memiller@mtu.edu (M.E.M.); lchavez@mtu.edu (L.L.B.-C.)
\*  Correspondence: spagnu39@msu.edu

**Abstract:** Grassland monitoring can be challenging because it is time-consuming and expensive to measure grass condition at large spatial scales. Remote sensing offers a time- and cost-effective method for mapping and monitoring grassland condition at both large spatial extents and fine temporal resolutions. Combinations of remotely sensed optical and radar imagery are particularly promising because together they can measure differences in moisture, structure, and reflectance among land cover types. We combined multi-date radar (PALSAR-2 and Sentinel-1) and optical (Sentinel-2) imagery with field data and visual interpretation of aerial imagery to classify land cover in the Masai Mara National Reserve, Kenya using machine learning (Random Forests). This study area comprises a diverse array of land cover types and changes over time due to seasonal changes in precipitation, seasonal movements of large herds of resident and migratory ungulates, fires, and livestock grazing. We classified twelve land cover types with user's and producer's accuracies ranging from 66%–100% and an overall accuracy of 86%. These methods were able to distinguish among short, medium, and tall grass cover at user's accuracies of 83%, 82%, and 85%, respectively. By yielding a highly accurate, fine-resolution map that distinguishes among grasses of different heights, this work not only outlines a viable method for future grassland mapping efforts but also will help inform local management decisions and research in the Masai Mara National Reserve.

**Keywords:** remote sensing; synthetic aperture radar; satellite imagery; grasslands; grass height; land cover

## 1. Introduction

Grasslands represent one of the Earth's most common vegetation types [1,2], covering nearly a fifth of the planet's land [3] and providing important ecological, economic, and cultural services. They are responsible for an estimated 16%–17% of global primary production [4–6], serve as hotspots for floral and faunal biodiversity [7,8], support endemic species [7–9], affect runoff and water quality [10], and contain up to 30% of the Earth's total soil carbon, thus reducing greenhouse gas emissions [2,8]. Grasslands are critical to the maintenance of human economies, livelihoods, and cultures, particularly for low-income and marginalized peoples [8,11]. In 2006, area allocated for livestock grazing covered a quarter of the Earth's ice-free land [12]. By supporting livestock, grasslands allow people to produce high-protein food, such as meat and milk [8], in addition to creating employment opportunities and generating income [11]. In recent decades, grasslands have suffered severe and increasing degradation;

between 2000 and 2010, approximately half of global grassland ecosystems underwent degradation, due primarily to climate change and human activities [13].

The importance of grasslands, paired with their vulnerability to climate change and human activity, makes their management a high priority. However, monitoring and managing grasslands is challenging. Field surveys of grass condition and production are costly and difficult to implement at the scale of large grasslands [14–17]. Remote sensing has great potential for improving grassland monitoring. While traditionally underutilized, remotely sensed data have been employed in several recent studies to detect grass cover, biomass, and height in temperate to hyper-arid grazing ecosystems (e.g., [17–25]). The studies cited here primarily used vegetation indices and fractions derived from data from passive, optical sensors, most commonly Landsat TM, to investigate correlates of spectral signatures with biophysical properties of grass, particularly biomass. Marsett et al. [21], Numata et al. [22], and Cimbelli and Vitale [25] specifically focused on predicting grass height, using Landsat [21,22] or Landsat and Sentinel [25] data to explain up to 80% of variation in grass height [21]. Only two of the studies cited here [24,25] used radar data in their analyses. Wang et al. [24] combined multitemporal optical data with multitemporal radar data collected in the X-band, C-band, and L-band; they described correlations between data collected via different sensors but did not incorporate field data on land cover. More recently, Cimbelli and Vitale [25] combined Landsat and Sentinel imagery with field data to predict grass height at medium resolution, but with limited success.

Although previous work suggests that remote sensing may be applicable to grassland management, several gaps in our current capabilities are apparent. First, most efforts to map grasslands fail to exploit the benefits of optical and radar sensor fusion. Second, most of the studies to date have been conducted using highly homogenous grasslands, such as pastures and prairies [17,19,22–24]. Saltz et al. [18], Schino et al. [20], Marsett et al. [21], and Cimbelli and Vitale [25], on the other hand, used spatially heterogeneous study sites, but with varying success in characterizing the biophysical properties of grass. The mapping algorithm applied to the most heterogeneous of the landscapes (a hyper-arid erosional cirque in Israel [18]) performed particularly poorly at characterizing plant cover.

Here, we seek to build upon previous work by extending the use of remote sensing in estimating land cover to include differentiation of discrete grass height classes in a dynamic savanna landscape representing a mosaic of open grasslands, shrubs, riverine forests, and wetlands. By integrating optical and radar imagery with a large field dataset, we aimed to produce a current and accurate land cover map of the Masai Mara National Reserve (henceforth, "the Reserve") in southwestern Kenya. This work is unique in three important ways. First, we aim to differentiate among grasses of different heights, which represent an important component of habitat suitability for various animals (e.g., large herbivores [26]), among other diverse land cover types in a heterogeneous landscape. Second, our methods apply a novel fusion of sensor imagery to the classification of land cover within a grassland ecosystem: PALSAR-2 radar imagery, Sentinel-1 radar imagery, and Sentinel-2 optical imagery. Third, our resulting land cover map provides a highly accurate, detailed, and novel map of a region that is of utmost conservation priority because it comprises the natural habitat of an enormously diverse fauna.

## 2. Background

We employed a combination of multi-date radar (PALSAR-2 and Sentinel-1) imagery and single date optical (Sentinel-2) imagery. Optical sensors are "passive," meaning they measure light (visible and infrared) emitted by the sun that has reflected off the Earth's surface. Synthetic aperture radar (SAR) sensors, on the other hand, are "active," meaning that they emit microwave energy and measure its backscatter from the Earth's surface [27]. Microwave energy, unlike visible and infrared light, penetrates cloud cover, affording radar sensors "all-weather" capability [28].

SAR imagery has been described as a perfect complement to optical imagery for several reasons [24]. First, as active sensors, SAR sensors can collect data at nighttime [27] and are not impeded by cloud cover. Second, radar and optical sensors collect data in different, complementary bands of electromagnetic

energy. Unlike reflected light measured by optical sensors, SAR backscatter is affected by standing water, soil inundation, surface roughness, and plant structure, biomass, and moisture content [24,27,29,30].

SAR sensors emit and detect microwave energy in the L-band (longest wavelength, lowest frequency), C-band, or X-band (shortest wavelength, highest frequency). The PALSAR-2 sensor measures backscatter in the L-band, meaning that it penetrates vegetative canopy and thus is sensitive to soil background, soil moisture, and standing water [26,31]. Co-polarized PALSAR-2 imagery is particularly useful for detection [30] and differentiation of wetland classes due to an enhanced double bounce effect from water surface and tree trunks [31]. Cross-polarized PALSAR-2 imagery, on the other hand, is sensitive to biomass, making it useful for distinction of woody from herbaceous vegetation [32]. The Sentinel-1 sensor measures backscatter in the C-band and provides information about vegetation structure and texture [33].

## 3. Methods

### 3.1. Study Area

The Reserve (1510 km$^2$ [34]) constitutes the northernmost portion of the Serengeti-Mara ecosystem. The Reserve consists primarily of open, rolling grassland with small patches of riparian vegetation along rivers and seasonal watercourses. Rainfall occurs bimodally, with most rain falling November–December and March–May [35]. The Reserve is bounded by the Oloololo Escarpment to the west and the Serengeti National Park to the south and is bisected north to south by the Mara River. The area west of the Mara River, known as the Mara Triangle, is managed by the Mara Conservancy, whereas the land east of the Mara River is managed by the Narok County Government. Due to differences in management, human disturbance, particularly livestock grazing, has been prevalent on the eastern side of the park in recent decades, whereas it rarely occurs on the western side.

In 2013, the Serengeti-Mara ecosystem was identified as one of only four remaining strongholds for carnivore conservation in East Africa [36]. It also seasonally hosts large herds of zebras and wildebeest migrating north from the Serengeti National Park [37] and southwest from the Loita Plains [38], and it is inhabited by many species of resident herbivores as well [37]. Altogether, the Masai Mara National Reserve supports 25% of Kenya's wildlife, based on estimates from the 1990s [39].

### 3.2. Remote Sensing Data

We combined multi-date imagery from the PALSAR-2, Sentinel-1, and Sentinel-2 sensors. Image dates were selected to be coincident with field data collection described in Section 3.3 below. All imagery was projected to the WGS 1984 Universal Transverse Mercator coordinate system, zone 36S. Images were stacked and clipped to the geometry of the Reserve boundary.

#### 3.2.1. ALOS-2 PALSAR-2 Radar Imagery

Radar imagery was collected by the PALSAR-2 sensor onboard the Advanced Land Observing Satellite 2 (ALOS-2) platform. PALSAR-2 (L-band, ~23 cm wavelength) images were recorded in Fine Beam Dual (FBD) mode, meaning that the sensor transmitted the signals horizontally and received them both horizontally (HH, known as co-polarization) and vertically (HV, known as cross-polarization). These data were collected at high resolution (10 × 10 m).

PALSAR-2 imagery was captured on two dates, 18 May 2018 and 13 July 2018. The imagery was collected in ascending orbit at 28.6° (all incident angles given apply at the center of the scene but vary across the extent of the scene). Only one frame was required to cover the entire extent of the study area. Images were calibrated to sigma-naught. We used a 3 × 3 median filter to account for speckle, the coherent addition of backscatter from multiple scatterers in the same resolution cell, which is inherent to all SAR imagery [27,28].

### 3.2.2. Sentinel Radar and Optical Imagery

Additional radar imagery was collected by the Sentinel-1 satellite constellation, operated by the European Space Agency (ESA). Sentinel-1 (C-band, ~5.5 cm wavelength) images were obtained in dual-polarization mode, meaning that signals were transmitted vertically and received both vertically (VV) and horizontally (VH). These data were collected in high-resolution mode (10 × 10 m).

Owing to the seasonal variation in the herbivore community composition, herbivore density, and rainfall, the Reserve is highly dynamic and surface features such as soil inundation and vegetative cover often change rapidly within a year. Therefore, it was critical to obtain satellite imagery and field data that were collected during the same time period. This guided our selection of Sentinel-1 data as the complement to the PALSAR-2 imagery. Sentinel-1 imagery was captured on two dates, 7 June 2018 and 19 June 2018. The imagery was collected in ascending orbit at an incident angle of ~33°. One frame was sufficient to cover the entire study area. Images were calibrated to sigma-naught and filtered using a 3 × 3 median filter.

Optical imagery was collected by the Sentinel-2B satellite, also operated by the ESA. Sentinel-2 data are collected in 13 spectral bands, ranging from ~443 nm – ~2190 nm. One frame of Sentinel-2 Level-1C top-of-atmosphere reflectance imagery was acquired for a single date, 11 June 2018. We planned to use imagery captured in July 2018 in order to include optical data coincident with the 13 July PALSAR-2 data, but an image collected early in the month was obscured by cloud cover, and subsequent image captures were collected while prescribed burns were occurring within the Reserve. We did not include the burned optical imagery in our analysis as our SAR imagery was collected before the burns, and our field verified sites did not include any already burned areas. Burned grasslands can experience enhanced regrowth and typically recover very rapidly. The visible and near-infrared bands (collected at 10 m resolution) along with vegetation red edge and shortwave infrared bands (collected at 20 m resolution) were used in this study. The bands collected at 20 m resolution were resampled to 10 m resolution using a nearest neighbor technique.

### 3.3. Training and Validation Data

Field data were collected throughout the Reserve between 4 June and 28 July 2018 to generate a supervised dataset for land cover classification (a blank field data collection sheet is available in the supplementary materials; Table S1). This time period did not overlap with either of the two rainy seasons and occurred prior to the arrival of the migratory herds of large herbivores. Therefore, grass height is unlikely to have changed substantially over the 54-day period of data collection. Our goal was to identify a minimum of six locations per land cover class (see Table 1 for definitions of land cover classes considered) to allow for a minimum of four training data and two validation data per class. We based our operations at the two field sites of the Mara Hyena Project (UTM coordinates: 751839 E, 9837939 N, and 724390 E, 9845214 N), and we therefore primarily collected data within the study areas monitored by the Mara Hyena Project. Specifically, we used ESRI ArcGIS to randomly generate 150 locations in the territories of three different hyena clans, covering a total of 71 km$^2$ west of the Mara River and a 61 km$^2$ area east of the Mara River. Random selection of locations was inefficient at identifying rare land cover classes, such as wetlands, water, barren ground, and *Acacia*-studded grassland (henceforth, shortened to "grass *Acacia*"). Therefore, we supplemented our field data by opportunistically sampling these rare land cover types when we encountered them in the field (this was also done by Bourgeau-Chavez et al. [27]).

At each field location, GPS (model: Garmin GPSMAP 78) coordinates were recorded using the averaging feature to improve horizontal accuracy and geotagged photographs were taken in the four cardinal directions and at nadir. For each sample area, we recorded the extent of the sample area, the land cover class, the dominant vegetation type, the approximate average height of the dominant vegetation, the percentage of vegetative cover, the distribution of the vegetation (homogeneous, heterogeneous, or patchy), and water inundation of the soil.

**Table 1.** Description of each land cover class mapped.

| Class | Description |
| --- | --- |
| Barren | Exposed light soil (sand), red soil (murram), dark soil (black cotton), and/or rock. Light soil is often exposed along rivers or dry creek beds or in transitional areas. Red soil is often exposed in murram quarries, on roads and airstrip runways, and in transitional areas. Dark soil is often exposed in overgrazed areas. |
| Riverine forest | Characterized by broadleaf evergreen trees and dead forests along rivers/streams. Woody vegetation must have a minimum height of four meters. |
| Upland forest | Characterized by broadleaf evergreen trees and dead forests occurring away (e.g., upland) from rivers/streams. Woody vegetation must have a minimum height of four meters. |
| Grass *Acacia* | *Acacia*-studded grasslands. Grass is the dominant vegetation type, followed by shrubs/trees of the genus *Acacia*. *Acacia* crown closure constitutes a minimum of 10% cover. |
| Grass *Balanites* | *Balanites*-studded grasslands. Grass is the dominant vegetation type, followed by *Balanites* trees. *Balanites* crown closure constitutes a minimum of 10% cover. |
| Tall grass | Grass plains where grass is 75 cm in height or taller. |
| Medium grass | Grass plains where grass is between 30 and 75 cm in height. |
| Short grass | Grass plains where grass is 30 cm in height or shorter. |
| Shrub | Patches of shrubs other than *Acacia*, typically dominated by shrubs of the genera *Croton* or *Euclea*. |
| Water | Areas persistently inundated in water that do not typically show annual drying out, such as streams, canals, rivers, lakes, estuaries, reservoirs, impounds, and bays. Water depth is typically 0.5 m or deeper, so surface and subsurface aquatic vegetation persistence is low. |
| Emergent wetland | Wetlands characterized by emergent or floating vegetation, including lily pads, cattails, sedges, and rushes. Some submergent vegetation may occur as well. The water table is at or near the earth's surface. Seasonal drying is variable within this class of wetlands. |
| Wet meadow | Wetland characterized primarily by inundated grasses and sedges along with some cattails and rushes. Following monsoons, the water table is at or near the earth's surface. Seasonal inundation and or drying are common phenomena. |

We collected field data at 233 locations. Polygons representing field data were hand-digitized using Google Earth Pro. Each polygon was drawn to include the GPS coordinates collected in the field. In some cases, these polygons were later reshaped to increase homogeneity within polygons to circumvent problems induced by spatial misalignment between sensors (Figure 1) and to avoid mixed pixel effects in dynamic areas. Some large original field site polygons were split to form two or more smaller, more homogeneous polygons in cases where the site was split by roads or the Mara River. A small number of polygons were deemed poor quality (e.g., not representative of a single land cover type, too heterogeneous) or were too small to avoid problems caused by mixed pixels or sensor misalignment and were therefore deleted. In total, we used 190 polygons based upon field observations. Of the field-derived training polygons, 136 were used for training and 54 were reserved for validation. Additionally, we added polygons for rare but easily detectable classes (e.g., water) from photo interpretation using both aerial imagery and our multi-sensor composite stack imagery; these points were not visited in the field. A total of 113 polygons were added based on visual interpretation of imagery. Of the 113 polygons added, 32 (28%) were upland forest, 20 (18%) were water, 13 (12%) were riverine forest, 8 (7%) were grass *Balanites*, 4 (4%) were grass *Acacia,* and the remaining 36 (32%) were wet meadow, emergent wetland, shrub, and barren. A total of 303 training and validation polygons collectively covering approximately 3.5 km$^2$ were used for the final classification and validation (Table 2).

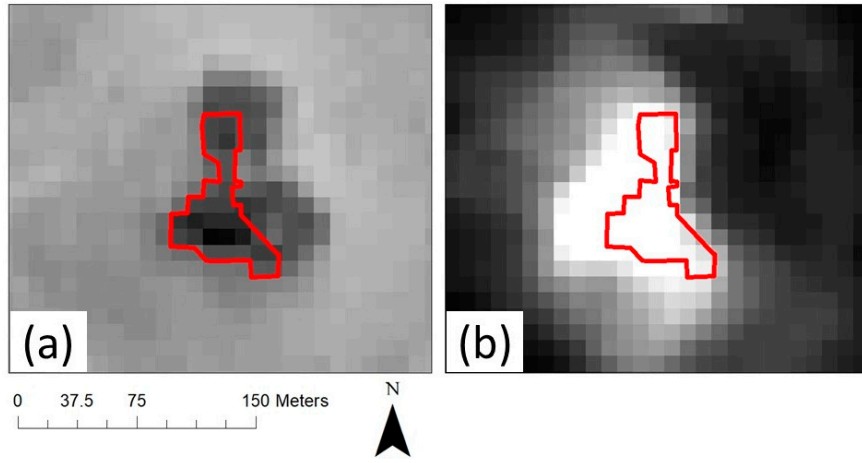

0    37.5    75          150 Meters

N

**Figure 1.** Great care had to be taken in delineating supervised data (polygons) to account for spatial misalignment of data collected by different sensors. Here, a patch of *Euclea* shrubs is shown using (**a**) Sentinel-2 band 4 (image captured 11 July 2018) and (**b**) Sentinel-1 cross-polarization (VH; image captured 13 July 2018). We aimed to limit the polygons to pixels where the patch, or feature, of interest, as shown by each sensor, overlapped.

**Table 2.** Summary of supervised data. Field data (FD) polygons were visited in research vehicles (ground-truthed). 190 of the original 233 FD polygons were included in the final training dataset. Polygons added via visual interpretation (VI) were not ground-truthed, but rather were identified via visual interpretation of remotely sensed imagery (PASLAR-2, Sentinel-1, and Sentinel-2) only.

| | Training Polygons | | | | | Validation Polygons | | | | |
|---|---|---|---|---|---|---|---|---|---|---|
| | FD | VI | Total | Pixels | Area (m$^2$) | FD | VI | Total | Pixels | Area (m$^2$) |
| Barren | 18 | 4 | 22 | 763 | 76,300 | 5 | 0 | 5 | 159 | 15,900 |
| Riverine forest | 6 | 13 | 19 | 2635 | 263,500 | 4 | 0 | 4 | 133 | 13,300 |
| Upland forest | 0 | 29 | 29 | 4161 | 416,100 | 5 | 3 | 8 | 727 | 72,700 |
| Grass *Acacia* | 5 | 4 | 9 | 250 | 25,000 | 2 | 0 | 2 | 50 | 5000 |
| Grass *Balanites* | 7 | 8 | 15 | 7303 | 730,300 | 3 | 0 | 3 | 1190 | 119,000 |
| Tall grass | 25 | 0 | 25 | 5462 | 546,200 | 6 | 0 | 6 | 959 | 95,900 |
| Medium grass | 30 | 0 | 30 | 2311 | 231,100 | 7 | 0 | 7 | 485 | 48,500 |
| Short grass | 18 | 0 | 18 | 1440 | 144,000 | 4 | 0 | 4 | 321 | 32,100 |
| Shrub | 18 | 7 | 25 | 1668 | 166,800 | 6 | 0 | 6 | 369 | 36,900 |
| Water | 4 | 20 | 24 | 799 | 79,900 | 5 | 0 | 5 | 170 | 17,000 |
| Emergent wetland | 1 | 11 | 12 | 1371 | 137,100 | 3 | 0 | 3 | 141 | 14,100 |
| Wet meadow | 4 | 14 | 18 | 1675 | 167,500 | 4 | 0 | 4 | 176 | 17,600 |
| Grand Total | 136 | 110 | 246 | 29,838 | 2,983,800 | 54 | 3 | 57 | 4880 | 488,000 |

### 3.4. Supervised Land Cover Classification

Land cover across the Reserve was classified using the process depicted in Figure 2. We first randomly partitioned our supervised data into two categories, a training set and a validation set. Polygons representing approximately 80% of the area for each class were included in the training set (Table 2), while polygons accounting for the remaining 20% were reserved as an independent validation set. Polygons representing sites that were visited during fieldwork were prioritized to be included in the validation data set. Supervised data added via visual interpretation of aerial imagery but not verified via ground-truthing were only used as validation data in cases where the field verified polygons did not reach the 20% threshold. This was done to ensure that validation used field-verified data whenever possible. In the final classified map, only the upland forest class contained polygons that were used as validation but were not visited in the field.

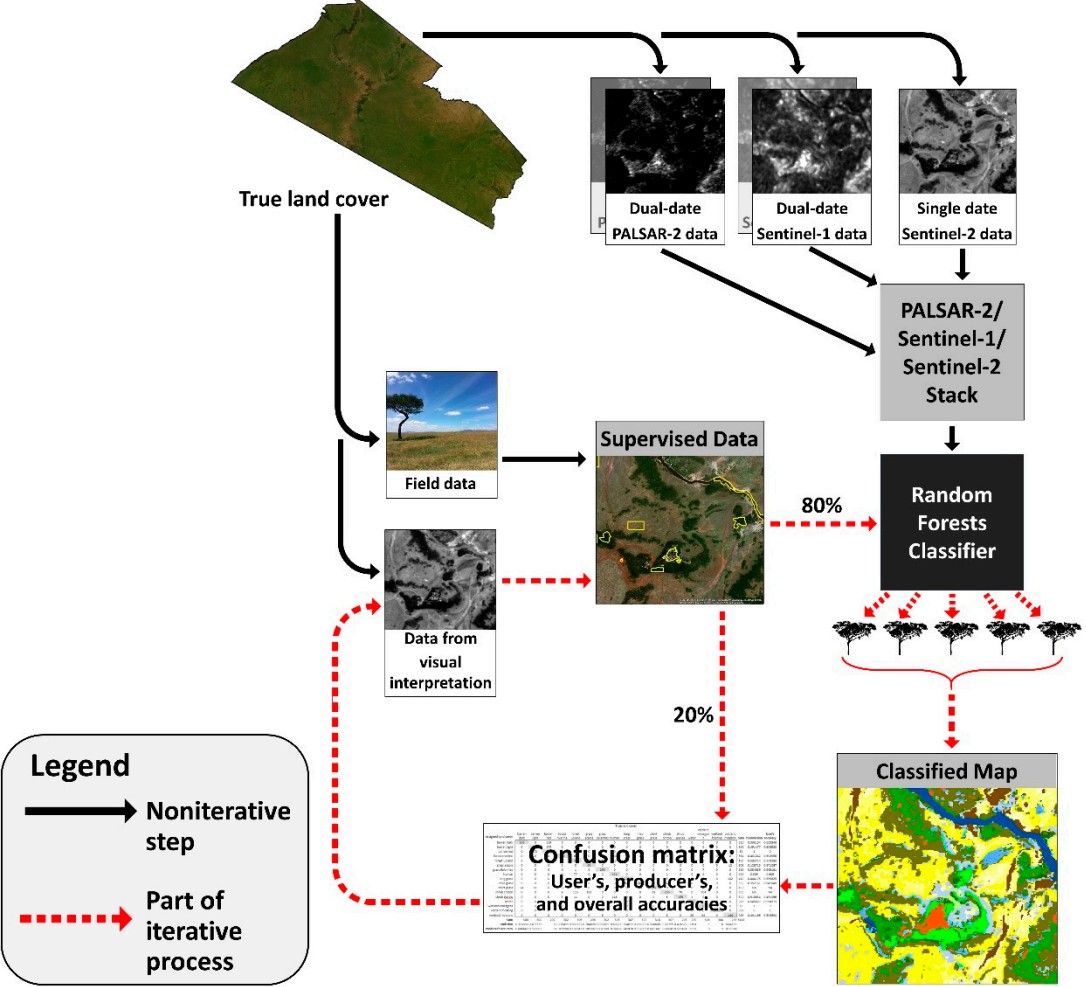

**Figure 2.** An overview of the land cover classification process. We reserved 20% of the polygons from the supervised dataset for validation, prioritizing field-verified sites. These validation polygons were not used in creating the Random Forests classifier. The other 80% of the supervised data and the multi-sensor composite stack imagery were input into a Random Forests classifier. The validation data were then used to construct a confusion matrix to assess accuracy. Visual inspection of the resultant land cover map and the confusion matrix informed subsequent refinement, addition, and deletion of supervised data and combination of classes.

The training dataset and imagery stack were used to predict land cover for each pixel in the study area using the machine learning algorithm Random Forests [40,41] in R version 3.6.1 [42]. The training data were selected from the training data polygons by selecting 100 random pixels from each land cover class. Random Forests uses both a random sample of the training data and a random subset of predictors (image bands) to create a decision tree that best classifies the data. This is repeated multiple times until a "forest" of decision trees is generated. Each decision tree generates a "vote" for the most likely land cover class for the given pixel, and the pixel is assigned to whichever land cover class receives the most votes. We used 500 trees in our classifier and used the default node size of one. The Random Forests classifier was deemed optimal for this study based on its high classification accuracy and relatively low processing time. Additional benefits of Random Forests include its insensitivity to missing data [29], such as pixels obscured by cloud cover [31] and nonpredictive input data, its capability for classifying datasets with many variables and relatively few training data [29,41], and the fact that it is easy to use [40] and allows for parallel processing [41].

After classification, the validation data were compared to the predictions of Random Forests to assess accuracy. The overall accuracy is the total number of correctly identified validation pixels divided by the total number of validation pixels. Within each land cover type, the producer's accuracy is the number of pixels correctly identified as the given land cover type divided by the total number of validation pixels of that class. In other words, we determined the percentage of pixels in a given class that were correctly identified. As producer's accuracy increases, omission error decreases. The user's accuracy is derived by dividing the number of pixels correctly identified as a given land cover type by the total number of pixels assigned to that class, whether they were correctly identified or not. In other words, for each given class, we determined the probability that a pixel identified as that class truly does belong to that class. User's accuracy, also known as reliability, increases as commission error decreases. Results are presented in an error, or confusion, matrix (Table 3). Our mapping goal was to achieve a minimum of 75% producer's and user's accuracies for each land cover type and an overall accuracy higher than 75%.

Based on the resultant map and confusion matrix, additional training polygons were added or removed via visual interpretation of high-resolution satellite imagery. A few field data had to be removed as the area of the land cover was too small to cover the minimum mapping area. Field-verified polygons that had been mapped using aerial imagery to define boundaries had to be buffered inward to avoid having mixed pixels within the scale of our 10 m resolution imagery. Additional riverine and upland forest polygon classes had to be added with visual interpretation of aerial imagery. After revision, data were once again assigned to training or validation data sets (prioritizing field-verified polygons for validation), and the Random Forests algorithm was run again. We aggregated the *Croton* shrub and *Euclea* shrub classes into a single shrub class.

Between iterations of Random Forests, if a class received additional training data (i.e., from visual interpretation), then the polygons in that class were once again randomly split. Therefore, for the classes requiring additional training polygons, validation data in the final classification may have been used in a prior run as training data. The most important classes of grass height were not affected, as we could not use visual interpretation. In the final classification iteration, we utilized a new image stack with different dates designed to avoid confusion with the burned areas. Given the change in predictor variables, we believe the final run can be considered independent of previous runs.

**Table 3.** Confusion (i.e., error) matrix for land cover classification of the Masai Mara National Reserve. Numbers represent pixels (10 × 10 m each). Numbers in gray cells represent pixels from the validation dataset that were correctly identified by the Random Forests classifier. Numbers falling outside the gray cells represent misclassified pixels.

| Classified Land Cover | True Land Cover | | | | | | | | | | | | Sum | Commission | User Acc. |
| | Barren | Riverine Forest | Upland Forest | Grass *Acacia* | Grass *Balanites* | Tall Grass | Medium Grass | Short Grass | Shrub | Water | Emergent Wetland | Wet Meadow | | | |
|---|---|---|---|---|---|---|---|---|---|---|---|---|---|---|---|
| Barren | 94 | 0 | 0 | 0 | 0 | 0 | 0 | 0 | 0 | 1 | 0 | 2 | 97 | 3% | 97% |
| Riverine forest | 0 | 79 | 7 | 1 | 0 | 0 | 0 | 0 | 0 | 0 | 1 | 1 | 89 | 11% | 89% |
| Upland forest | 0 | 10 | 87 | 0 | 0 | 0 | 0 | 0 | 2 | 0 | 0 | 0 | 99 | 12% | 88% |
| Grass *Acacia* | 0 | 0 | 0 | 82 | 5 | 1 | 0 | 0 | 10 | 0 | 1 | 9 | 108 | 24% | 76% |
| Grass *Balanites* | 0 | 0 | 0 | 0 | 94 | 1 | 0 | 0 | 0 | 0 | 0 | 0 | 95 | 1% | 99% |
| Tall grass | 0 | 0 | 0 | 4 | 1 | 82 | 7 | 2 | 1 | 0 | 0 | 0 | 97 | 15% | 85% |
| Medium grass | 0 | 0 | 0 | 2 | 0 | 7 | 88 | 10 | 0 | 0 | 0 | 0 | 107 | 18% | 82% |
| Short grass | 5 | 0 | 0 | 0 | 0 | 7 | 6 | 87 | 0 | 0 | 0 | 0 | 105 | 17% | 83% |
| Shrub | 0 | 12 | 15 | 0 | 0 | 1 | 0 | 0 | 81 | 0 | 13 | 0 | 122 | 34% | 66% |
| Water | 0 | 0 | 0 | 0 | 0 | 0 | 0 | 0 | 0 | 102 | 0 | 0 | 102 | 0% | 100% |
| Emergent wetland | 0 | 4 | 0 | 6 | 0 | 0 | 0 | 0 | 0 | 0 | 83 | 3 | 96 | 14% | 86% |
| Wet meadow | 0 | 0 | 0 | 8 | 0 | 0 | 0 | 0 | 0 | 0 | 3 | 83 | 94 | 12% | 88% |
| Sum | 99 | 105 | 109 | 103 | 100 | 99 | 101 | 99 | 94 | 103 | 101 | 98 | | | |
| Omission | 5% | 25% | 20% | 20% | 6% | 17% | 13% | 12% | 14% | 1% | 18% | 15% | | | |
| Prod. Acc. | 95% | 75% | 80% | 80% | 94% | 83% | 87% | 88% | 86% | 99% | 82% | 85% | | | 86% |

## 4. Results

We combined a 10 m resolution imagery stack (dual date PALSAR-2, dual date Sentinel-1, and single date Sentinel-2) with training and validation data to assign land cover of the Masai Mara National Reserve using a Random Forests classifier. The resultant map (Figure 3) had an overall accuracy of 86%. The producer's accuracies for individual land cover classes ranged from 75% to 100% and the user's accuracies ranged from 66% to 100% (Table 3). This map will be made publicly and freely available as a Tagged Image File Format, compatible with ArcGIS and QGIS, via the Michigan Tech Research Institute (Ann Arbor, MI, USA) website.

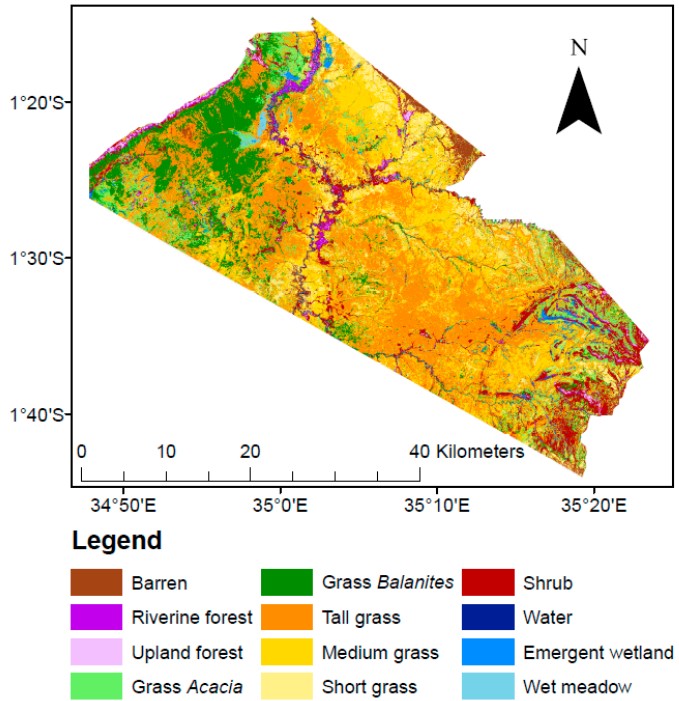

**Figure 3.** Land cover map of the Masai Mara National Reserve, with an overall accuracy of 86%.

Recall that the producer's accuracy is the likelihood with which a validation pixel for a given class was correctly classified, whereas the user's accuracy is the likelihood that a pixel assigned to a given class truly represents that class. The highest class-level accuracy was obtained for water, such as permanent water holes and rivers, which had producer's and user's accuracies of 99% and100%, respectively. Most of the observed confusion occurred between riverine forest, upland forest, shrub, and grass *Acacia*, but this error was not substantial; this confusion increased class-level omission error by a maximum of 21% (riverine forest) and commission error by a maximum of 22% (shrub). Not surprisingly, medium grass had the lowest user's accuracy (82%) of the grass heights due to confusion between tall and short grasses. That is, slightly more confusion occurred between tall and medium grass and between medium and short grass, than between tall and short grass. The three grass height land covers were all identified with accuracies of 82% or higher, with accuracies as high as 85% (user's) and 88% (producer's) obtained (Table 3).

Variable importance of our image bands was quantified using the mean decrease in accuracy metric (Figure 4) [41]. We looked at the importance of all the image bands to the overall classification, and we also looked at the importance of bands to classifying the three grass height land cover types. Values represent the loss in out-of-bag classification accuracy for each input band if that band had been excluded or permutated. Cross-polarized backscatter from L-band PALSAR-2 collected on 13 July 2018 was the most important band for the full classification as well as for the grass classes individually. The cross-polarized component of radar backscatter is typically due to significant volume scattering and

is especially sensitive to biomass, so the importance of the variable for discriminating grass height, shrub, and forest is not surprising. Sentinel-2 red edge bands were also important, which highlights the strength of those bands for distinguishing vegetation types. Of some interest is the relatively low importance of C-band Sentinel-1 data. Cimbelli and Vitale [25] also found Sentinel-1 to have limited value in assessing grass height in a study region in Italy.

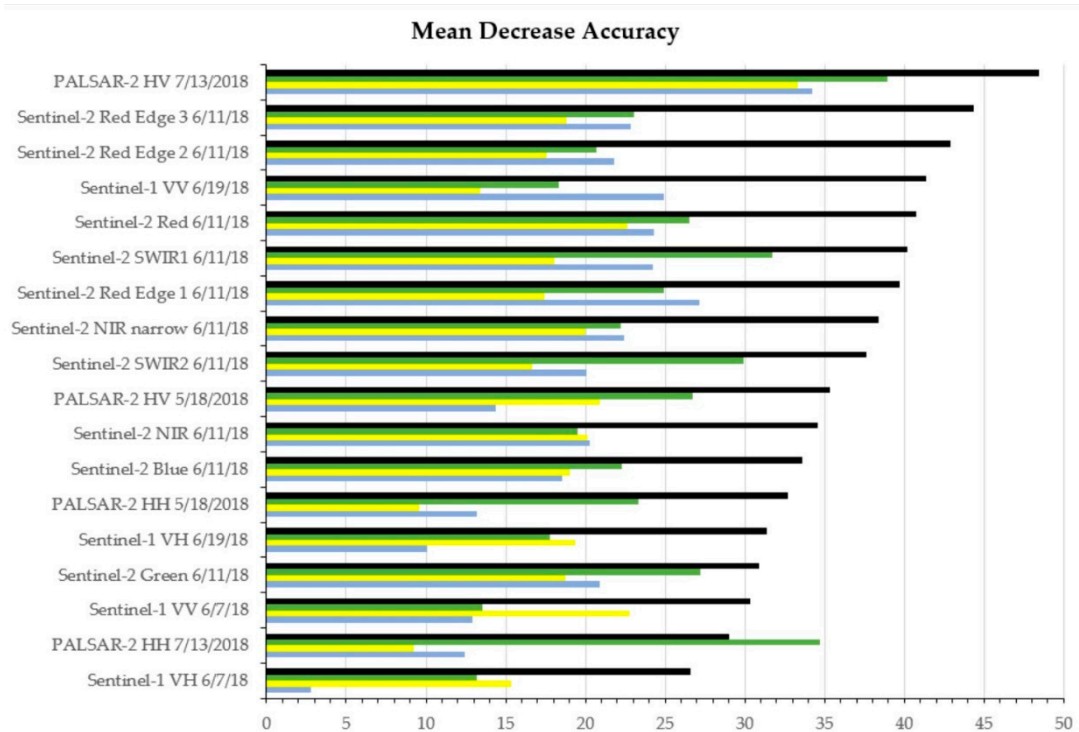

**Figure 4.** Mean Decrease Accuracy for all classes (black), short grass (green), medium grass (yellow), and tall grass (blue).

The most common land cover across the Reserve was tall grass (31.0%), followed by medium grass (19.7%). Short grass covered 10.19% of the study area, similar to grass *Acacia* (12.0%), grass *Balanites* (11.0%), and shrub (8.8%) (Table 4). Collectively, grass constituted 60.9% of the entire study area. The rarest classes were water (0.3%), followed by wetlands (1.3% emergent wetland and 1.1% wet meadow), and forests (1.1% riverine forest and 0.9% upland forest). Visual inspection of the map (Figure 3) corroborates our personal observations that upland forest primarily occurs along the Olololo Escarpment, which runs along the western boundary of the Reserve, atop inselbergs, and sometimes on hills, particularly those with human settlements, including tourist lodges. Riverine forest is found along the Mara River, with some also appearing along the Talek River and other small rivers. Shrubs were over four times as widespread as riverine and upland forest combined. Also consistent with our personal observations, *Balanites* trees appeared to occur more densely west of the Mara River than east, specifically immediately east of the Olololo Escarpment (Figure 3).

Wetlands occur infrequently throughout this habitat, comprising only 2.4% of the entire Reserve. Many of the existing wetlands (especially wet meadow) occurred west of the Mara River, in the Mara Triangle. This too is consistent with our personal observations.

**Table 4.** Total area and percentage of Masai Mara National Reserve covered by each land cover class, based on the output of Random Forests.

| Land Cover | Total Area (km$^2$) | Percentage of Study Area |
|---|---|---|
| Barren | 45 | 3% |
| Riverine forest | 18 | 1% |
| Upland forest | 14 | 1% |
| Grass *Acacia* | 191 | 12% |
| Grass *Balanites* | 176 | 11% |
| Tall grass | 496 | 31% |
| Medium grass | 315 | 20% |
| Short grass | 163 | 10% |
| Shrub | 141 | 9% |
| Water | 4 | < 1% |
| Emergent wetland | 20 | 1% |
| Wet meadow | 17 | 1% |

## 5. Discussion

Our land cover classification method proved highly effective in this heterogeneous and temporally dynamic ecosystem. Excitingly, this method was successful in differentiating grasses of different heights, which, to our knowledge, has not been previously achieved in such a diverse landscape mosaic. A mapping technique with this capability is particularly important for an area like the Masai Mara National Reserve, which is composed primarily of grasslands (83.8% of the area is covered by short, medium, and tall grass as well as grass *Acacia* and grass *Balanites*).

Visual inspection of our map (Figure 3) suggests several differences between the Mara Triangle (west of the Mara River), which is managed by the Mara Conservancy, and the portion of the Reserve managed by the Narok County government (east of the Mara River). Some of these differences may be due to differences in management, i.e., active vs. passive management of livestock grazing within the Reserve. Within the eastern side, progression from south to north (e.g., towards the northern boundary of the Reserve) coincides with a transition from tall to medium to short grass and sometimes to large patches of barren ground. That is, grasslands in this region appear to diminish with increasing proximity to the northern boundary. The southern boundary of the Reserve is the Kenyan/Tanzanian border, beyond which lies the Serengeti National Park. Beyond the northern boundary, on the other hand, some land is protected whereas other areas are not. For instance, the communities of Talek and N'Tipiliguani, which lie immediately north of the Reserve, have developed rapidly, leading to a fivefold increase in illegal livestock grazing in the park between 2008 and 2015 [43,44]. This could potentially explain the south–north transition in grass height and cover.

However, some differences in land cover between the western and eastern sides of the park are more likely attributable to naturally occurring topographic variation. For example, the Mara Triangle seems to contain higher proportions of wetlands than the rest of the Reserve. This may in part be attributed to the higher rainfall the Mara Triangle receives compared to the east side of the Reserve due to local precipitation patterns created by the Lake Victoria convergence zone [45]. Additionally, the Mara Triangle is more densely populated by *Acacia* and *Balanites* trees than is the area east of the Mara River.

Although we consider this landscape to be a savanna–woodland mosaic, it is worth noting that grass is far more common than woody vegetation. Open grasslands constitute the majority of the entire Reserve (60.9%), followed by grasslands studded with sparse *Acacia* and *Balanites* (23.0%), and then shrubs (8.8%). Forests constitute only 2.0% of land cover within the Reserve. Historically, these grasslands have been maintained by frequent fire disturbance and uprooting of woody vegetation by elephants [46]. The frequent resetting of the successional clock by elephants and fires would explain why trees are less common than shrubs, which in turn are less common than grasses. Riverine forest is found primarily along rivers, particularly the Mara River. Upland forest is primarily distributed along

the top of the Oloololo Escarpment but is also commonly found atop inselbergs. Inselbergs, which are hills or small mountains that rise abruptly, typically consisting of granite or gneiss rock [47] (Figure 5), in the Reserve are typically topped by patches of shrubs or trees (Figure 6). Generally, the vegetation on inselbergs is distinct from that of the surrounding land cover due to harsh edaphic (i.e., amount of soil cover) and microclimatic (i.e., evaporation rate and degree of insolation) conditions [47]. Thus, inselbergs clearly contribute to the Reserve's diversity of land cover and vegetation types.

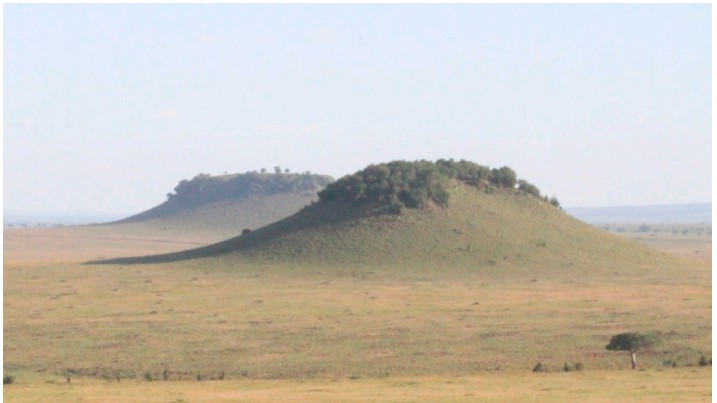

**Figure 5.** Inselbergs typically occur as abruptly rising, solitary, monolithic hills or mountains. They are particularly common in the tropics and subtropics [47]. This photo was taken in the southern Mara Triangle, where inselbergs are quite common.

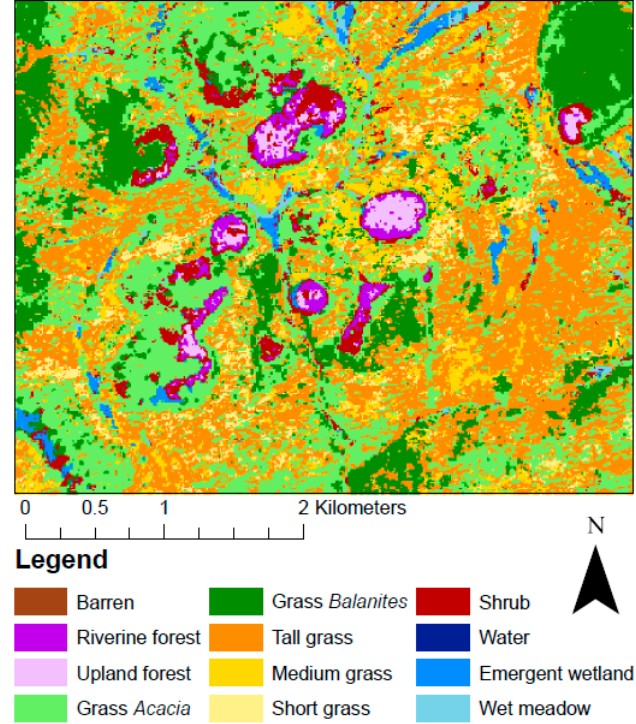

**Figure 6.** A cluster of inselbergs in the southwestern Mara Triangle, on which rock and dense woody vegetation occur.

Although our overall and class-level accuracies were consistently high, the confusion between shrub and tree cover is worth mentioning. That is, riverine forest, upland forest, grass *Acacia*, and shrub were sometimes confused, resulting in a 5% decrease in the overall accuracy (Table 3). In future applications, the addition of lidar data (aerial or space-based) to measure vegetation height has great

potential to help differentiate forests and shrubs and perhaps even grass heights. The inclusion of a texture metric might also resolve some of this confusion. Grass *Acacia* and grass *Balanites* are composed of flat grasslands with sparse tall trees and thus should be very rough-textured. Forests consist of trees (canopy layer) and shrubs (understory) of various heights, so they should also be somewhat rough in texture. Shrub patches, on the other hand, are relatively homogeneous in height and therefore should be relatively smooth in texture.

We hope that this map will prove useful for ecological research within the Reserve. The Reserve is home to a plethora of research projects, focusing on spotted hyenas (e.g., [48]), lions (e.g., [49]), cheetahs (e.g., [50]), baboons (e.g., [51]), Martial Eagles (e.g., [52]), ungulates (e.g., [44]), and river ecology (e.g., [53]), to name a few. It should also be useful to managers in the Masai Mara. Overgrazing by livestock represents a serious and ongoing threat to this ecosystem. This map may inform management decisions by identifying large patches of barren ground or short grass, which may represent problem areas warranting allocation of management efforts. Furthermore, application of our methods using optical and radar imagery collected over time may represent a highly accurate, feasible, and cost-effective method for monitoring grassland condition.

These methods should also be applicable to other savanna–woodland landscapes in East Africa. They have proven effective, despite the spatially heterogeneous and temporally dynamic nature of this ecosystem. Furthermore, they are capable of differentiating short, medium, and tall grass. Open grasslands comprise well over half of this study area, so differentiating grass height is important for delivering a detailed, informative land cover map.

There are several notable advantages afforded by these methods. First, the combination of PALSAR-2, Sentinel-1, and Sentinel-2 yields data across a wider range of the electromagnetic spectrum [26,27,29,30] than radar or optical imagery could separately, which provides more unique spectral signatures corresponding to different land cover types. The diversity of bands in which radiation is measured allows for inference about diverse features, from soil inundation to vegetation canopy structure [27,28]. Second, the use of multidate data yields valuable information, particularly in a temporally dynamic landscape such as the Masai Mara National Reserve. For example, wet meadows are prone to seasonal drying, and therefore single-date imagery could falsely classify wet meadow as grassland or permanent water. Incorporation of imagery from a second date can reduce errors in classification of seasonally dynamic land cover classes. Third, the Random Forests classifier is well adapted for this application, due to its high classification accuracy, insensitivity to missing data [29], ability to function with relatively few training data [29,41], ease of use, and low processing time [40]. Finally, using an independent validation data set allows for accuracy measurement and identification of sources of confusion.

Additional work is needed to incorporate the temporal effects of wildfire on grass heights. We know a portion of our map changed very rapidly during our field observation period due to fire. None of our field data were gathered after burning, but some training sites did burn after our field visits. Therefore, we chose to map the vegetation before the fire occurred, but it is of great importance to remember the product is a snapshot of conditions over a relatively short temporal period.

This method should also be applicable in classifying land cover in mixed grassland systems beyond East Africa. Remote sensing may represent a highly effective, logistically feasible, affordable method to monitor grassland conditions, which can inform effective management. The method developed and assessed in this paper demonstrates the utility of remotely sensed imagery in differentiating grass height, even within a spatially diverse and temporally dynamic ecosystem. Future work should seek to expand the application of this method to other mixed grassland ecosystems, explore imagery sources which are more accessible, and test the feasibility of developing algorithms that are not as heavily reliant on expensive field observations. If algorithms do require field data, it would be useful to determine how much data will be needed and for what temporal scales the maps are useful.

## 6. Conclusions

This study represents the first study to our knowledge to use remotely sensed data to accurately classify discrete classes of grass height among other diverse land cover types in a tropical savanna-woodland landscape mosaic, which is highly spatially heterogeneous and temporally dynamic. We combined multi-date radar imagery (PALSAR-2 and Sentinel-1) and optical imagery (Sentinel-2) from a single season with training data from ground-truthing (i.e., field data) and image interpretation to assign land cover at a fine spatial resolution ($10 \times 10$ m) using a machine learning algorithm, Random Forests [40]. We reserved 20% of our supervised data for validation, allowing us to assess and report user's, producer's, and overall accuracy and subsequently add and refine supervised data to improve classification in an iterative process.

The resultant map was highly accurate, achieving user's and producer's accuracies ranging from 66%–100% and an overall map accuracy of 86%. In addition to classifying a wide variety of land cover types, from open water to forests, we were able to distinguish discrete grass heights (short, medium, and tall) with user's accuracies of 83%, 82%, and 85%, respectively. Furthermore, confusion between grassland and non-grassland land cover occurred at low rates of 1% (omission) and 4% (commission). Overall, most of the confusion in classification occurred between shrubs, forests, and grasslands dotted with *Acacia* trees or shrubs. However, confusion was still low.

We expect that this fine-resolution, highly accurate land cover map of an ecologically important protected area will inform wildlife managers and allow researchers to address new questions regarding habitat preference and land cover change over time. Furthermore, these methods can be repeated or expanded upon for implementation in other mixed grassland ecosystems.

**Supplementary Materials:** The following are available online at http://www.mdpi.com/2072-4292/12/7/1086/s1, Table S1: blank field data sheet.

**Author Contributions:** Conceptualization, O.S.B.S., J.C.J., and Z.M.L.; methodology, M.J.B., Z.M.L., and M.E.M.; field data collection, O.S.B.S., J.C.J., and Z.M.L.; software, M.J.B. and M.E.M.; validation, M.J.B. and M.E.M.; formal analysis, O.S.B.S. and M.J.B.; resources, K.E.H. and L.L.B.-C.; data curation, O.S.B.S., J.C.J., M.J.B., Z.M.L., and M.E.M.; writing—original draft preparation, O.S.B.S.; writing—review and editing, O.S.B.S., J.C.J., M.J.B., Z.M.L., M.E.M., K.E.H., and L.L.B.-C.; visualization, O.S.B.S., J.C.J., and M.J.B.; project administration, O.S.B.S. and Z.M.L.; funding acquisition, O.S.B.S., J.C.J., M.J.B., Z.M.L., M.E.M., K.E.H., and L.L.B.-C. All authors have read and agreed to the published version of the manuscript.

**Funding:** This work was supported by NSF grants OISE1853934 and IOS1755089 to K.E.H. Ground truthing was made possible by travel funding from the Ecology, Evolutionary Biology, and Behavior (EEBB) program (EEBB Summer Fellowship and EEBB Travel Fellowship) and the Graduate School (Research Enhancement Award) at Michigan State University. The PALSAR-2 imagery was provided through Grant # APS3120 to Laura Bourgeau-Chavez, under the Japan Aerospace Exploration Agency's 6th Research Announcement for the Advanced Land Observing Satellite-2 (ALOS-2). Remote sensing training was made possible by MichiganView through AmericaView grant #AV18-MI-01 to MTRI.

**Acknowledgments:** We thank the Kenyan National Commission for Science, Technology and Innovation, the Kenya Wildlife Service, the Narok County Government, the Mara Conservancy, Brian Heath, James Sindiyo, and the Naibosho Conservancy for permissions to conduct this research.

**Conflicts of Interest:** The authors declare no conflict of interest. The funders had no role in the design of the study; in the collection, analyses, or interpretation of data; in the writing of the manuscript, or in the decision to publish the results.

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
