# Peer review of "Mapping Kenyan Grassland Heights Across Large Spatial Scales with Combined Optical and Radar Satellite Imagery"

_remotesensing, doi:10.3390/rs12071086_

Round 1

Reviewer 1 Report

Overall a very well written and put together piece. The science was sound and the detail provided was perfect; especially for anyone wanting to reproduce the efforts in their own system. Including avenues of improvement and furthering accuracy in the models at the end was also much appreciated. I did have some thoughts on some minor points that I will include below:

Line 39-40 needs citations

Line 56 include what was used to explain 80% of height variation

Line 63 include citations for papers that are referred to

line 79 remove the word "an"

Figure 1 line colors are hard differentiate to in greyscale. Consider using broken and solid lines instead

Figure 3 first line remove truly

Reviewer 2 Report

The paper addresses land classification using optical and radar imagery together with supervised machine learning.

The paper is well written and organized and is organized properly, with a good introduction and background presentation.

General Comments

I have a major issue with the methodology.

If you’re trying to classify land according to certain existing classes, I was expecting a supervised learning classification approach but it seems you presented a mixed approach, where the validation step is performed through Visual Inspection and Polygon Revision based on quality measures (overall, producer and user accuracies).

However, I’m not sure I fully grasped your classification methodology:

. Do you use Random Forests (RF) only prior to validation?

. How do you exactly perform the validation step?

Nevertheless, you have never used a test step with data not used previously, so you can’t really tell if your method works on a different area (even with the same characteristics) because your solution may be biased towards the data used on the model’s development.

Other Comments

Why did you find important to included the failed attempt of imagery capture? (lines 156-157)

Are the time differences for some tasks important regarding land composition? (data/images gathered from May through August).

The different steps of the classification method need to be clarified:

. Training set vs Imagery stack

. Is the spectral signature obtained pixelwise?

. What do you mean by remaining pixels (line 211) and what are the similarities used for class assignment?

. How do you ensure some class continuity (if the method is pixelwise)?

. Are the classes used balanced?

. I think the differences between the Producer and User accuracies are not clear in the text.

. How do you revise, delete and add polygons on the validation step? Manually?

. Are the additions made on top of the previous optimized polygons?

How do you assess the effectiveness of your method concerning the temporal dynamics of the ecosystem?

Please check the text for a couple typos.

Reviewer 3 Report

This study nicely explains a methodology for mapping grassland heights combining multi-date SAR data and optical data. The authors have evaluated the feasibility of integrating optical and SAR data to differentiate among three classes of grasses with different heights and to produce an accurate land cover map. Although the authors claim for the novelty of the “fusion of sensor imagery for classification”, the use of SAR and optical data for classification have been used for years by several authors, e.g. Navarro et al., 2016. The experiment is well described and explained, in particular the data used, the in situ data collection and the methodology. Also the relevant issues related with the land cover classification and the park management are well described in the discussing section. The manuscript deserves to be published, but I have some comments on two things concerning the methodology, listed below, that authors could address and correct before its final submission.

Comments:

As far as I understood the field work were between June and August 2018 and the image acquisition were between 18 May 2018 and 13 July 2018. I suppose that within this period the grass has grown some centimetres. How much confident are you about the discrimination between the three classes of height. Looking at the confusion matrix, the user’s accuracy is less for the medium grass than for the other two. This issue should be discussed in the paper. Random Forest Algorithm (line 256). You are mentioning that the “random forest algorithm was repeated until acceptable accuracy was achieved and that you made a limited number of changes were made”. I suggest to detail the changes you have made in each iteration and the parameters you have used in the Random Forest algorithm (The number of trees in the forest, The maximum depth of the tree, etc. I am concerned about the possibility of editing the in situ data, in the limit you can edit the field data (deleting and inserting new data) until you reach the 100% accuracy. It is very unlikely that the in situ data are incorrect since it was collected by your team. It would be very interesting to identify among the imagery used in the classification which one has more impact on the classification. I would like to see importance of each variable (bands and images). How many variables are you considering in the Random Forest algorithm and how many samples? This should be clearly stated. Figure 4. Please insert the label of the Mara river. It is not necessary to have the coordinate labels around the map, could be only on the bottom and left. Please reduce the label to degree and minutes. Also remove the label “1 centimeter = 4 kilometers”, it not correct, and also in figure 6 In the first paragraph of the conclusion you are mentioning that this “is the first study to use remote sensing data to accurately predict classes of grass height”. Your study is a good study concerning the determination of the grass height but there are many studies concerning the determination of the rice height using SAR data and also for grass height (Cimbelli, A. and Vitale, V. (2017) Grassland Height Assessment by Satellite Images. Advances in Remote Sensing, 6, 40-53.) I suggest to change accordingly the conclusion’ first paragraph.

Round 2

Reviewer 2 Report

The authors addressed all comments and changed / improved the text accordingly.

Now it is clear that the methodology is a supervised two-step equivalent procedures with the second one aiming to refine the results from the first one.

Nevertheless I still have one comment left:

The use of validation is probably used here in a different way than some readers might expect.

The validation set is usually used to improve/fine tune the method’s parameters after training and there is usually a third set for independent testing (with samples unknown to the model). But in your text the two concepts seem mingled.

In your approach you use the validation set for the usual purposes but then, on the second step after enhancement of the polygons you use it again and no independent testing is really performed. I understand that you sampled again both sets - training & validation -, at random (in a cross-validation way), resulting in different sets than the ones on the first classification step. However, even if you used different sets’ composition, the samples were known to the model beforehand (from the previous classification) and the model’s results may be somehow biased.

Thus, don’t you think there should be a final truly independent test dataset to evaluate the model’s performance or, at least, acknowledge this limitation on the results?
